# Evolutionary Population Curriculum for Scaling Multi-Agent Reinforcement Learning

**Qian Long**[*]
CMU
qianlong@cs.cmu.edu

**Zihan Zhou**[*]
SJTU
footoredo@sjtu.edu.cn

**Abhibav Gupta**
CMU, Facebook AI Research
abhinavg@cs.cmu.edu

**Fei Fang**
CMU
feif@cs.cmu.edu

**Yi Wu**[†]
OpenAI
jxwuyi@openai.com

**Xiaolong Wang**[†]
UCSD
xiw012@ucsd.edu

## Abstract

In multi-agent games, the complexity of the environment can grow exponentially as the number of agents increases, so it is particularly challenging to learn good policies when the agent population is large. In this paper, we introduce *Evolutionary Population Curriculum* (EPC), a curriculum learning paradigm that scales up Multi-Agent Reinforcement Learning (MARL) by progressively increasing the population of training agents in a stage-wise manner. Furthermore, EPC uses an evolutionary approach to fix an objective misalignment issue throughout the curriculum: agents successfully trained in an early stage with a small population are not necessarily the best candidates for adapting to later stages with scaled populations. Concretely, EPC maintains multiple sets of agents in each stage, performs mix-and-match and fine-tuning over these sets and promotes the sets of agents with the best adaptability to the next stage. We implement EPC on a popular MARL algorithm, MADDPG, and empirically show that our approach consistently outperforms baselines by a large margin as the number of agents grows exponentially. The source code and videos can be found at https://sites.google.com/view/epciclr2020/.

## 1 Introduction

Most real-world problems involve interactions between multiple agents and the problem becomes significantly harder when there exist complex cooperation and competition among agents. Inspired by the tremendous success of deep reinforcement learning (RL) in single-agent applications, such as Atari games (Mnih et al., 2013), robotics manipulation (Levine et al., 2016), and navigation (Zhu et al., 2017; Wu et al., 2018; Yang et al., 2019), it has become a popular trend to apply deep RL techniques into multi-agent applications, including communication (Foerster et al., 2016; Sukhbaatar et al., 2016; Mordatch & Abbeel, 2018), traffic light control (Wu et al., 2017), physical combats (Bansal et al., 2018), and video games (Liu et al., 2019; OpenAI, 2018).

A fundamental challenge for multi-agent reinforcement learning (MARL) is that, as the number of agents increases, the problem becomes significantly more complex and the variance of policy gradients can grow exponentially (Lowe et al., 2017). Despite the advances on tackling this challenge via actor-critic methods (Lowe et al., 2017; Foerster et al., 2018), which utilize decentralized actors and centralized critics to stabilize training, recent works still scale poorly and are mostly restricted to less than a dozen agents. However, many real-world applications involve a moderately large population of agents, such as algorithmic trading (Wellman et al., 2005), sport team competition (Hausknecht & Stone, 2015), and humanitarian assistance and disaster response (Meier, 2015), where one agent should collaborate and/or compete with all other agents. When directly applying the existing MARL algorithms to complex games with a large number of agents, as we will show in Sec. 5.3, the agents may fail to learn good strategies and end up with little interaction with other agents even when collaboration is significantly beneficial. Yang et al. (2018) proposed a provably-converged mean-field formulation to scale up the actor-critic framework by feeding the state information and the *average value* of nearby agents' actions to the critic. However, this formulation strongly relies on the assumption that the value function for each agent can be *well approximated* by the mean of local

---

[*]Equal contribution
[†]Equal advising

pairwise interactions. This assumption often does not hold when the interactions between agents become complex, leading to a significant drop in the performance.

In this paper, we propose a general learning paradigm called *Evolutionary Population Curriculum* (EPC), which allows us to scale up the number of agents exponentially. The core idea of EPC is to progressively increase the population of agents throughout the training process. Particularly, we divide the learning procedure into multiple stages with increasing number of agents in the environment. The agents first learn to play in simpler scenarios with less agents and then leverage these experiences to gradually adapt to later stages with more agents and ultimately our desired population.

There are two key components in our curriculum learning paradigm. To process the varying number of agents during the curriculum procedure, the policy/critic needs to be *population-invariant*. So, we choose a self-attention (Vaswani et al., 2017) based architecture which can generalize to an arbitrary number of agents with a fixed number of parameters. More importantly, we introduce an *evolutionary selection* process, which helps address the misalignment of learning goals across stages and improves the agents' performance in the target environment. Intuitively, our within-stage MARL training objective only incentivizes agents to overfit a particular population in the current stage. When moving towards a new stage with a larger population, the successfully trained agents may not adapt well to the scaled environment. To mitigate this issue, we maintain multiple sets of agents in each stage, evolve them through cross-set mix-and-match and parallel MARL fine-tuning in the scaled environment, and select those with better adaptability to the next stage.

EPC is RL-algorithm agnostic and can be potentially integrated with most existing MARL algorithms. In this paper, we illustrate the empirical benefits of EPC by implementing it on a popular MARL algorithm, MADDPG (Lowe et al., 2017), and experimenting on three challenging environments, including a predator-prey-style individual survival game, a mixed cooperative-and-competitive battle game, and a fully cooperative food collection game. We show that EPC outperforms baseline approaches by a large margin on all these environments as the number of agents grows even exponentially. We also demonstrate that our method can improve the stability of the training procedure.

## 2 Related Work

**Multi-Agent Reinforcement Learning:** It has been a long history in applying RL to multi-agent games (Littman, 1994; Shoham et al., 2003; Panait & Luke, 2005; Wright et al., 2019). Recently, deep RL techniques have been applied into the multi-agent scenarios to solve complex Markov games and great algorithmic advances have been achieved. Foerster et al. (2016) and He et al. (2016) explored a multi-agent variant of deep Q-learning; Peng et al. (2017) studied a fully centralized actor-critic variant; Foerster et al. (2018) developed a decentralized multi-agent policy gradient algorithm with a centralized baseline; Lowe et al. (2017) proposes the MADDPG algorithm which extended DDPG to the multi-agent setting with decentralized policies and centralized Q functions. Our population curriculum approach is a general framework for scaling MARL which can be potentially combined with any of these algorithms. Particularly, we implement our method on top of the MADDPG algorithm in this paper and take different MADDPG variants as baselines in experiments. There are also other works studying large-scale MARL recently (Lin et al., 2018; Jiang & Lu, 2018; Yang et al., 2018; Suarez et al., 2019), which typically simplify the problem by weight sharing and taking only local observations. We consider a much more general setting with global observations and unshared-weight agents. Additionally, our approach is a general learning paradigm which is complementary to the specific techniques proposed in these works.

**Attention-Based Policy Architecture:** Attention mechanism is widely used in RL policy representation to capture object level information (Duan et al., 2017; Wang et al., 2018), represent relations (Zambaldi et al., 2018; Malysheva et al., 2018; Yang et al., 2019) and extract communication channels (Jiang & Lu, 2018). Iqbal & Sha (2019) use an attention-based critic. In our work, we utilize an attention module in both policy and critic, inspired by the transformer architecture (Vaswani et al., 2017), for the purpose of generalization to an arbitrary number of input entities.

**Curriculum Learning:** Curriculum learning can be tracked back to Elman (1993), and its core idea is to "start small": learn the easier aspects of the task first and then gradually increase the task difficulty. It has been extended to deep neural networks on both vision and language tasks (Bengio et al., 2009) and much beyond: Karras et al. (2017) propose to progressively increase the network capacity for synthesizing high quality images; Murali et al. (2018) apply a curriculum over the control space for robotic manipulation tasks; several works (Wu & Tian, 2016; Florensa et al., 2017; Sukhbaatar et al., 2017; Wang et al., 2019) have proposed to first train RL agents on easier goals

and switch to harder ones later. Baker et al. (2019) show that multi-agent self-play can also lead to autocurricula in open-ended environments. In our paper, we propose to progressively increase the number of the agents as a curriculum for better scaling multi-agent reinforcement learning.

**Evolutionary Learning:** Evolutionary algorithms, originally inspired by Darwin's natural selection, has a long history (Bäck & Schwefel, 1993), which trains a population of agents in parallel, and let them evolve via crossover, mutation and selection processes. Recently, evolutionary algorithms have been applied to learn deep RL policies with various aims, such as to enhance training scalability (Salimans et al., 2017), to tune hyper-parameters (Jaderberg et al., 2017), to evolve intrinsic dense rewards (Jaderberg et al., 2018), to learn a neural loss for better generalization (Houthooft et al., 2018), to obtain diverse samples for faster off-policy learning (Khadka & Tumer, 2018), and to encourage exploration (Conti et al., 2018). Leveraging this insight, we apply evolutionary learning to *better scale MARL*: we train several groups of agents in each curriculum stage and keep evolving them to larger populations for the purpose of better adaptation towards the desired population scale and improved training stability. Czarnecki et al. (2018) proposed a similar evolutionary mix-and-match training paradigm to progressively increase agent capacity, i.e., larger action spaces and more parameters. Their work considers a fixed environment with an increasingly more complex agent and utilizes the traditional parameter crossover and mutation during evolution. By contrast, we focus on scaling MARL, namely an increasingly more complex environment with a growing number of agents. More importantly, we utilize MARL fine-tuning as an implicit mutation operator rather than the classical way of mutating parameters, which is more efficient, guided and applicable to even a very small number of evolution individuals. A similar idea of using learning for mutation is also considered by Gangwani & Peng (2018) in the single-agent setting.

## 3 BACKGROUND

**Markov Games:** We consider a multi-agent Markov decision processes (MDPs) (Littman, 1994). Such an $N$-agent Markov game is defined by state space $\mathcal{S}$ of the game, action spaces $\mathcal{A}_1, ..., \mathcal{A}_N$ and observation spaces $\mathcal{O}_1, ..., \mathcal{O}_N$ for each agent. Each agent $i$ receives a private observation correlated with the state $\mathbf{o}_i : \mathcal{S} \mapsto \mathcal{O}_i$ and produces an action by a stochastic policy $\boldsymbol{\pi}_{\theta_i} : \mathcal{O}_i \times \mathcal{A}_i \mapsto [0, 1]$ parameterized by $\theta_i$. Then the next states are produced according to the transition function $\mathcal{T} : \mathcal{S} \times \mathcal{A}_1 \times ... \times \mathcal{A}_N \mapsto \mathcal{S}$. The initial state is determined by a distribution $\rho : \mathcal{S} \mapsto [0, 1]$. Each agent $i$ obtains rewards as a function of the state and its action $r_i : \mathcal{S} \times \mathcal{A}_i \mapsto \mathbb{R}$, and aims to maximize its own expected return $R_i = \sum_{t=0}^{T} \gamma^t r_i^t(s^t, a_i^t)$, where $\gamma$ is a discount factor and $T$ is the time horizon. To minimize notation, we omit subscript of policy when there is no ambiguity.

**Multi-Agent Deep Deterministic Policy Gradient (MADDPG):** MADDPG (Lowe et al., 2017) is a multi-agent variant of the deterministic policy gradient algorithm (Silver et al., 2014). It learns a centralized Q function for each agent which conditions on global state information to resolve the non-stationary issue. Consider $N$ agents with deterministic policies $\boldsymbol{\mu} = \{\boldsymbol{\mu}_1, ..., \boldsymbol{\mu}_N\}$ where $\boldsymbol{\mu}_i : \mathcal{O}_i \mapsto \mathcal{A}_i$ is parameterized by $\theta_i$. The policy gradient for agent $i$ is:

$$\nabla_{\theta_i} J(\theta_i) = \mathbb{E}_{\mathbf{x}, a \sim \mathcal{D}}[\nabla_{\theta_i} \boldsymbol{\mu}_i(o_i) \nabla_{a_i} Q_i^{\boldsymbol{\mu}}(\mathbf{x}, a_1, ..., a_N)|_{a_i = \boldsymbol{\mu}_i(o_i)}], \tag{1}$$

Here $\mathcal{D}$ denotes the replay buffer while $Q_i^{\boldsymbol{\mu}}(\mathbf{x}, a_1, ..., a_N)$ is a *centralized action-value function* for agent $i$ that takes the actions of all agents, $a_1, ..., a_N$ and the state information $\mathbf{x}$ (i.e., $\mathbf{x} = (o_1, ..., o_N)$ or simply $\mathbf{x} = s$ if $s$ is available). Let $\mathbf{x}'$ denote the next state from the environment transition. The replay buffer $\mathcal{D}$ contains experiences in the form of tuples $(\mathbf{x}, \mathbf{x}', a_1, ..., a_N, r_1, ..., r_N)$. Suppose the centralized critic $Q_i^{\boldsymbol{\mu}}$ is parameterized by $\phi_i$. Then it is updated via:

$$\mathcal{L}(\phi_i) = \mathbb{E}_{\mathbf{x}, a, r, \mathbf{x}'}[(Q_i^{\boldsymbol{\mu}}(\mathbf{x}, a_1, ..., a_N) - y)^2], \quad y = r_i + \gamma \, Q_i^{\boldsymbol{\mu}'}(\mathbf{x}', a_1', ..., a_N')\big|_{a_j' = \boldsymbol{\mu}_j'(o_j)}, \tag{2}$$

where $\boldsymbol{\mu}' = \{\boldsymbol{\mu}_{\theta_1'}, ..., \boldsymbol{\mu}_{\theta_N'}\}$ is the set of target policies with delayed parameters $\theta_i'$. Note that the centralized critic is only used during training. At execution time, each policy $\boldsymbol{\mu}_{\theta_i}$ remains decentralized and only takes local observation $o_i$.

## 4 EVOLUTIONARY POPULATION CURRICULUM

In this section, we will first describe the base network architecture with the self-attention mechanism (Vaswani et al., 2017) which allows us to incorporate a flexible number of agents during training. Then we will introduce the population curriculum paradigm and the evolutionary selection process.

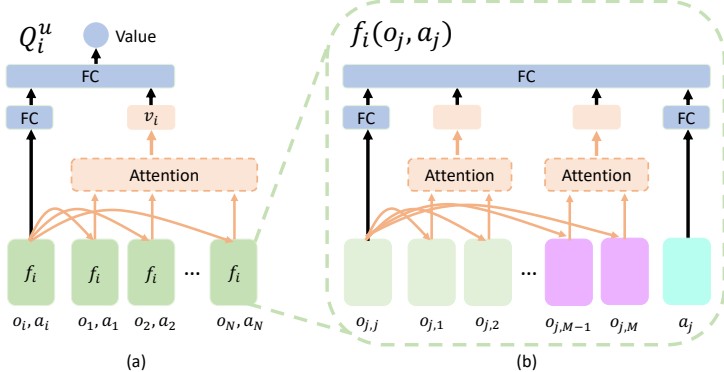

Figure 1: Our population-invariant Q function: (a) utilizes the attention mechanism to combine embeddings from different observation-action encoder $f_i$; (b) is a detailed description for $f_i$, which also utilizes an attention module to combine $M$ different entities in one observation.

## 4.1 POPULATION-INVARIANT ARCHITECTURE

We describe our choice of architecture based on the MADDPG algorithm (Lowe et al., 2017), which is population-invariant in the sense that both the Q function and the policy can take in an arbitrary number of input entities. We first introduce the Q function (Fig. 1) and then the policy.

We adopt the decentralized execution framework, so each agent has its own Q function and policy network. Particularly for agent $i$, its centralized Q function is represented as follows:

$$Q_i^{\boldsymbol{\mu}}(\mathbf{x}, a_1, \ldots, a_N) = h_i([g_i(f_i(o_i, a_i)), v_i]), \text{ where } v_i = \text{attention}(f_i(o_j, a_j) \forall j \neq i) \quad (3)$$

Here $f_i(o_j, a_j)$ is an *observation-action encoder* (the green box in Fig. 1(a)) which takes in the observation $o_j$ and the action $a_j$ from agent $j$, and outputs the agent embedding of agent $j$; $v_i$ denotes the *global attention embedding* (the orange box in Fig. 1(a)) over all the agent embeddings. We will explain $v_i$ and $f_i$ later. $g_i$ is a 1-layer fully connected network processing the embedding of the $i$th agent's own observation and action. $h_i$ is a 2-layer fully connected network that takes the concatenation of the output of $g_i$ and the global attention embedding $v_i$ and outputs the final Q value.

**Attention Embedding $v_i$:** We define the attention embedding $v_i$ by a weighted sum of each agent's embedding $f_i(o_j, a_j)$ for $j \neq i$:

$$v_i = \sum_{j \neq i} \alpha_{i,j} f_i(o_j, a_j) \quad (4)$$

The coefficient $\alpha_{i,j}$ is computed by

$$\alpha_{i,j} = \frac{\exp(\beta_{i,j})}{\sum_{j \neq i} \exp(\beta_{i,j})}, \quad \beta_{i,j} = f_i^T(o_i, a_i) W_\psi^T W_\phi f_i(o_j, a_j) \quad (5)$$

where $W_\psi$ and $W_\phi$ are parameters to learn. $\beta_{i,j}$ computes the correlation between the embeddings of agent $i$ and every other agent $j$ via an inner product. $\alpha_{i,j}$ is then obtained by normalizing $\beta_{i,j}$ by a softmax function. Since we represent the observations and actions of other agents with a weighted mean $v_i$ from Eq. 4, we can model the interactions between agent $i$ and an arbitrary number of other agents, which allows us to easily increase the number of agents in our curriculum training paradigm.

**Observation-Action Encoder $f_i$:** We now define the structure of $f_i(o_j, a_j)$ (Fig. 1(b)). Note that the observation of agent $j$, $o_j$, also includes many entities, i.e., states of all visible agents and objects in the game. Suppose $o_j$ contains $M$ entities, i.e., $o_j = [o_{j,1}, \ldots, o_{j,M}]$. $M$ may also vary as the agent population scales over training procedure or simply during an episode when some agents die. Thus, we apply another attention module to combine these entity observations together in a similar way to how $v_i$ is computed (Eq. 4, 5).

In more details, we first apply an entity encoder for each entity type to obtain *entity embeddings* of all the entities within that type. For example, in $o_j$, we can have embeddings for agent entities (green boxes in Fig. 1(b)) and landmark/object entities (purple boxes in Fig. 1(b)). Then we apply an attention module over each entity type by attending the entity embedding of agent $j$ to all the entities of this type to obtain an attended *type embedding* (the orange box in Fig. 1(b)). Next, we concatenate

all the type embeddings together with the entity embedding of agent $j$ as well as its action embedding. Finally, this concatenated vector is forwarded to a fully connected layer to generate the output of $f_i(o_j, a_j)$. Note that in the overall critic network of agent $i$, the same encoder $f_i$ is applied to every observation-action pair so that the network can maintain a fixed size of parameters even when the number of agents increases significantly.

**Policy Network:** The policy network $\boldsymbol{\mu}_i(o_i)$ has a similar structure as the observation-action encoder $f_i(o_i, a_i)$, which uses an attention module over the entities of each type in the observation $o_i$ to adapt to the changing population during training. The only difference in this network is that the action $a_i$ is not included in the input. Notably, we do not share parameters between the Q function and the policy.

## 4.2 POPULATION CURRICULUM

We propose to progressively scale the number of agents in MARL with a curriculum. Before combining with the evolutionary selection process, we first introduce a simpler version, the vanilla population curriculum (PC), where we perform the following stage-wise procedure: (i) the initial stage starts with MARL training over a small number of agents using MADDPG and our population-invariant architecture; (ii) we start a new stage and double[1] the number of agents by cloning each of the existing agents; (iii) apply MADDPG training on this scaled population until convergence; (iv) if the desired number of agents is not reached, go back to step (ii).

Mathematically, given $N$ trained agents with parameters $\boldsymbol{\theta} = \{\theta_1, ..., \theta_N\}$ from the previous stage, we want to increase the number of the agents to $2N$ with new parameters $\tilde{\boldsymbol{\theta}} = \{\tilde{\theta}_1, ..., \tilde{\theta}_N, ..., \tilde{\theta}_{2N}\}$ for the next stage . In this vanilla version of population curriculum, we simply initialize $\tilde{\boldsymbol{\theta}}$ by setting $\tilde{\theta}_i \leftarrow \theta_i$ and $\tilde{\theta}_{N+i} \leftarrow \theta_i$, and then continue MADDPG training on $\tilde{\boldsymbol{\theta}}$ to get the final policies for the new stage. Although $\tilde{\theta}_i$ and $\tilde{\theta}_{N+i}$ are both initialized from $\theta_i$, as training proceeds, they will converge to different policies since these policies are trained in a decentralized manner in MADDPG.

## 4.3 EVOLUTIONARY SELECTION

Introducing new agents by directly cloning existing ones from the previous stage has a clear limitation: the policy parameters suitable for the previous environment are not necessarily the best initialization for the current stage as the population is scaled up. In the purpose of better performance in the final game with our desired population, we need to promote agents with better adaptation abilities during early stages of training.

Therefore, we propose an evolutionary selection process to facilitate the agents' scaling adaption ability during the curriculum procedure. Instead of training a single set of agents, we maintain $K$ parallel sets of agents in each stage, and perform crossover, mutation and selection among them for the next stage. This is the last piece in our proposed *Evolutionary Population Curriculum* (EPC) paradigm, which is essentially population curriculum enhanced by the evolutionary selection process.

Specifically, we assume the agents in the multi-agent game have $\Omega$ different roles. Agents in the same role have the same action set and reward structure. For example, we have $\Omega = 2$ roles in a predator-prey game, namely predators and prey, and $\Omega = 1$ role of agents for a fully cooperative game with homogeneous agents. For notation conciseness, we assume there are $N_1$ agents of role 1, namely $A_1 = \{\boldsymbol{\mu}_1, ..., \boldsymbol{\mu}_{N_1}\}$; $N_2$ agents of role 2, namely $A_2 = \{\boldsymbol{\mu}_{N_1+1}, ..., \boldsymbol{\mu}_{N_1+N_2}\}$, and so on. In each stage, we keep $K$ parallel sets for each role of agents, denoted by $A_i^{(1)}, ..., A_i^{(K)}$ for role $i$, and take a 3-step procedure, i.e., *mix-and-match (crossover)*, *MARL fine-tuning (mutation)* and *selection*, as follows to evolve these $K$ parallel sets of agents for the next stage.

**Mix-and-Match (Crossover):** In the beginning of a curriculum stage, we scale the population of agents from $N$ to $2N$. Note that we have $K$ parallel agent sets of size $N_i$ for role $i$, namely $A_i^{(1)}, ..., A_i^{(K)}$. We first perform a mix-and-match over these parallel sets *within* every role $i$: for each set $A_i^{(j)}$, we pair it with all the $K$ sets of the same role, which leads to $K(K+1)/2$ new scaled agent sets of size $2N_i$. Given these scaled sets of agents, we then perform another mix-and-match *across* all the $\Omega$ roles: we pick one scaled set for each role and combine these $\Omega$ selected sets to produce a scaled game with $2N$ agents. For example, in the case of $\Omega = 2$, we can pick one agent set $A_1^{(k_1)}$ from the first role and another agent set $A_2^{(k_2)}$ from the second role to form a scaled game.

---

[1]Generally, we can scale up the population with any constant factor by introducing any amount of cloned agents. We use the factor of 2 as a concrete example here for easier understanding.

Thus, there are $C_{\max} = (K(K+1)/2)^\Omega$ different combinations in total through this mix-and-match process. We sample $C$ games from these combinations for mutation in the next step. Since we are mixing parallel sets of agents, this process can be considered as the *crossover* operator in standard evolutionary algorithms.

**MARL Fine-Tuning (Mutation):** In standard evolutionary algorithms, mutations are directly performed on the parameters, which is inefficient in high-dimensional spaces and typically requires a large amount of mutants to achieve sufficient diversity for evolution. Instead, here we adopt MARL fine-tuning in each curriculum stage (step (iii) in vanilla PC) as our guided mutation operator, which naturally and efficiently explores effective directions in the parameter space. Meanwhile, due to the training variance, MARL also introduces randomness which benefits the overall diversity of the evolutionary process. Concretely, we apply parallel MADDPG training on each of the $C$ scaled games generated from the mix-and-match step and obtain $C$ mutated sets of agents for each role.

**Selection:** Among these $C$ mutated sets of agents for each role, only the best $K$ mutants can survive. In the case of $\Omega = 1$, the fitness score of a set of agents is computed as their average reward after MARL training. In other cases when $\Omega \geq 2$, given a particular mutated set of agents of a specific role, we randomly generate games for this set of agents and other mutated sets from different agent roles. We take its average reward from these randomly generated games as the fitness score for this mutated set. We pick the top-$K$ scored sets of agents in each role to advance to the next curriculum stage.

---

**Algorithm 1:** Evolutionary Population Curriculum

---

**Data:** environment $E(N, \{A_i\}_{1 \leq i \leq \Omega})$ with $N$ agents of $\Omega$ roles, desired population $N_d$, initial population $N_0$, evolution size $K$, mix-and-match size $C$

**Result:** a set of $N_d$ best policies

$N \leftarrow N_0$;

initialize $K$ parallel agent sets $A_i^{(1)}, \ldots, A_i^{(K)}$ for each role $1 \leq i \leq \Omega$;

initial parallel MARL training on $K$ games, $E(N, \{A_i^{(j)}\}_{1 \leq i \leq \Omega})$ for $1 \leq j \leq K$;

**while** $N < N_d$ **do**
    $N \leftarrow 2 \times N$;
    **for** $1 \leq j \leq C$ **do**
        for each role $1 \leq i \leq \Omega$: $j_1, j_2 \leftarrow \mathrm{unif}(1, K)$, $\tilde{A}_i^{(j)} \leftarrow A_i^{(j_1)} + A_i^{(j_2)}$ (*mix-and-match*);

    MARL training in parallel on $E(N, \{\tilde{A}_i^{(j)}\}_{1 \leq i \leq \Omega})$ for $1 \leq j \leq C$ (*guided mutation*) ;
    **for** *role* $1 \leq i \leq \Omega$ **do**
        **for** $1 \leq j \leq C$ **do**
            $S_i^{(j)} \leftarrow \mathbb{E}_{k_{t \neq i} \sim [1, C]} \left[ \text{avg. rewards on } E(N, \{\tilde{A}_1^{(k_1)}, \ldots, \tilde{A}_i^{(j)}, \ldots, \tilde{A}_\Omega^{(k_\Omega)}\}) \right]$
            (*fitness*);
        $A_i^{(1)}, \ldots, A_i^{(K)} \leftarrow$ top-$K$ w.r.t. $S_i$ from $\tilde{A}_i^{(1)}, \ldots, \tilde{A}_i^{(C)}$ (*selection*);

**return** the best set of agents in each role, i.e., $\{A_i^{(k_i^\star)} | k_i^\star \in [1, K] \; \forall 1 \leq i \leq \Omega\}$;

---

**Overall Algorithm:** Finally, when the desired population is achieved, we take the best set of agents in each role based on their last fitness scores as the output. We conclude the detailed steps of EPC in Alg. 1. Note that in the first curriculum stage, we just train $K$ parallel games without mix-and-match or mutation. So, EPC simply selects the best from the $K$ initial sets in the first stage while the evolutionary selection process only takes effect starting from the second stage. We emphasize that although we evolve multiple sets of agents in each stage, the three operators, mix-and-match, MARL fine-tuning and selection, are all **perfectly parallel**. Thus, the evolutionary selection process only introduces little influence on the overall training time. Lastly, EPC is an RL-algorithm-agnostic learning paradigm that can be potentially integrated with any MARL algorithm other than MADDPG.

## 5 EXPERIMENT

We experiment on three challenging environments, including a predatory-prey-style *Grassland* game, a mixed-cooperative-and-competitive *Adversarial Battle* game and a fully cooperative *Food Collection* game. We compare EPC with multiple baseline methods on these environments with different scales of agent populations and show consistently large gains over the baselines. In the following, we will first introduce the environments and the baselines, and then both qualitative and quantitative performances of different methods on all three environments.

## 5.1 Environments

All these environments are built on top of the particle-world environment (Mordatch & Abbeel, 2018) where agents take actions in discrete timesteps in a continous 2D world.

***Grassland***: In this game, we have $\Omega = 2$ roles of agents, $N_S$ sheep and $N_W$ wolves, where sheep moves twice as fast as wolves. We also have a fixed amount of $L$ grass pellets (food for sheep) as green landmarks (Fig. 2a). A wolf will be rewarded when it collides with (eats) a sheep,

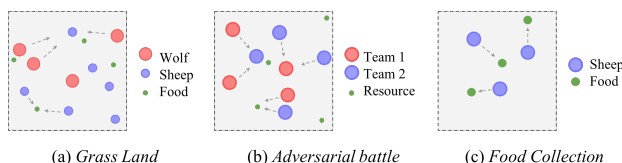

(a) *Grass Land*    (b) *Adversarial battle*    (c) *Food Collection*

Figure 2: Environment Visualizations

and the (eaten) sheep will obtain a negative reward and becomes inactive (dead). A sheep will be rewarded when it comes across a grass pellet and the grass will be collected and respawned in another random position. Note that in this survival game, each individual agent has its own reward and does not share rewards with others.

***Adversarial Battle***: This scenario consists of $L$ units of resources as green landmarks and two teams of agents (i.e., $\Omega = 2$ for each team) competing for the resources (Fig. 2b). Both teams have the same number of agents ($N_1 = N_2$). When an agent collects a unit of resource, the resource will be respawned and *all the agents in its team* will receive a positive reward. Furthermore, if there are more than *two* agents from team 1 collide with *one* agent from team 2, *the whole team 1* will be rewarded while the trapped agent from team 2 will be deactivated (dead) and the whole team 2 will be penalized, and vice versa.

***Food Collection***: This game has $N$ food locations and $N$ fully cooperative agents ($\Omega = 1$). The agents need to collaboratively occupy as many food locations as possible within the game horizon (Fig. 2c). Whenever a food is occupied by any agent, the whole team will get a reward of $6/N$ in that timestep for that food. The more food occupied, the more rewards the team will collect.

In addition, we introduce collision penalties as well as auxiliary shaped rewards for each agent in each game for easier training. All the environments are fully observable so that each agent needs to handle a lot of entities and react w.r.t. the global state. More environment details are in Appx. A.

## 5.2 Methods and Metric

We evaluate the following approaches in our experiments: (1) the MADDPG algorithm (Lowe et al., 2017) with its original architecture (**MADDPG**); (2) the provably-converged mean-field algorithm (Yang et al., 2018) (**mean-field**); (3) the MADDPG algorithm with our population-invariant architecture (**Att-MADDPG**); (4) the vanilla population curriculum without evolutionary selection (**vanilla-PC**); and (5) our proposed EPC approach (**EPC**). For EPC parameters, we choose $K = 2$ for *Grassland* and *Adversarial Battle* and $K = 3$ for *Food Collection*; for the mix-and-match size $C$, we simply set it $C_{\max}$ and enumerate all possible mix-and-match combinations instead of random sampling. All the baseline methods are trained until the same amount of accumulative episodes as EPC took. More training details can be found in Appx. B.

For *Grassland* and *Adversarial Battle* with $\Omega = 2$, we evaluate the performance of different methods by competing their trained agents against our EPC trained agents. Specifically, in *Grassland*, we let sheep trained by each approach compete with the wolves from EPC and collect the average sheep reward as the evaluation metric for sheep. Similarly, we take the same measurement for wolves from each method. In *Adversarial Battle*, since two teams are symmetric, we just evaluate the shared reward of one team trained by each baseline against another team by EPC as the metric. For *Food Collection* with $\Omega = 1$, since it is fully cooperative, we take the team reward for each method as the evaluation metric. In addition, for better visualization, we plot the *normalized scores* by normalizing the rewards of different methods between 0 and 1 in each scale for each game. More evaluation details are in Appx. C.

## 5.3 Qualitative Results

In *Grassland*, as the number of wolves goes up, it becomes increasingly more challenging for sheep to survive; meanwhile, as the sheep become more intelligent, the wolves will be incentivized to be more aggressive accordingly. In Fig. 3, we illustrate two representative matches for competition, including one using the MADDPG sheep against the EPC wolves (Fig. 3a), and the other between the EPC sheep and the MADDPG wolves (Fig. 3b). From Fig. 3a, we can observe that the MADDPG sheep can be easily eaten up by the EPC wolves (note that dark circle means the sheep is eaten). On

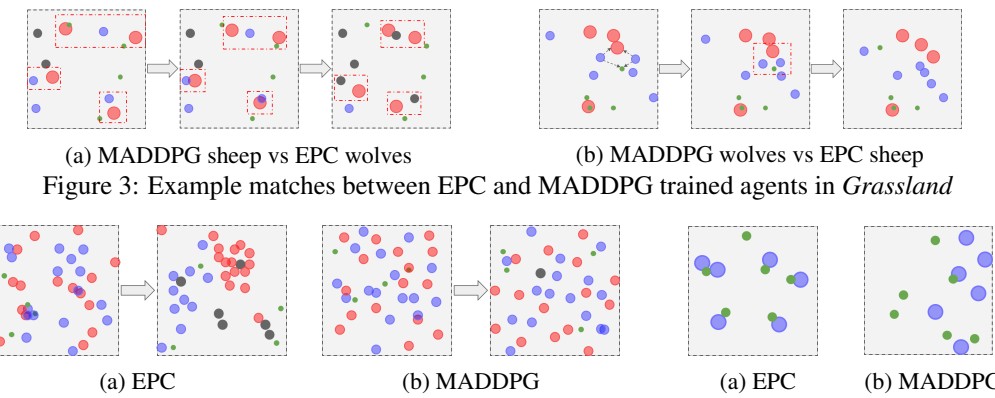

(a) MADDPG sheep vs EPC wolves     (b) MADDPG wolves vs EPC sheep

Figure 3: Example matches between EPC and MADDPG trained agents in *Grassland*

(a) EPC                    (b) MADDPG                    (a) EPC          (b) MADDPG

Figure 4: *Adversarial Battle*: dark particles are dead agents.          Figure 5: *Food Collection*

the other hand, in Fig. 3b, we can see that the EPC sheep learns to eat the grass and avoid the wolves at the same time.

In *Adversarial Battle*, we visualize two matches in Fig. 4 with one over agents by EPC (Fig. 4a) and the other over agents by MADDPG (Fig. 4b). We can clearly see the collaborations between the EPC agents: although the agents are initially spread over the environment, they learn to quickly gather as a group to protect themselves from being killed. While for the MADDPG agents, their behavior shows little incentives to cooperate or compete — these agents stay in their local regions throughout the episode and only collect resources or kill enemies very infrequently.

In *Food Collection* (Fig. 5), the EPC agents in Fig. 5a learn to spread out and occupy as many food as possible to maximize the team rewards. While only one agent among the MADDPG agents in Fig. 5b successfully occupies a food in the episode.

### 5.4 QUANTITATIVE RESULTS

**Quantitative Results in *Grassland***

In the *Grassland* game, we perform curriculum training by starting with 3 sheep and 2 wolves, and gradually increase the population of agents. We denote a game with $N_S$ sheep and $N_W$ wolves by "scale $N_S$-$N_W$". We start with scale 3-2 and gradually increase the game size to scales 6-4, 12-8 and finally 24-16. For the two curriculum learning approach, vanilla-PC and EPC, we train over $10^5$ episodes in the first curriculum stage (scale 3-2) and fine-tune the agents with $5 \times 10^4$ episodes after mix-and-match in each of the following stage. For other methods that train the agents from scratch, we take the same accumulative training iterations as the curriculum methods for a fair comparison.

**Main Results:** We report the performance of different methods for each game scale in Fig. 6a. Overall, there are little differences between the mean-field approach and the original MADDPG algorithm while the using the population-invariant architecture (i.e., Att-MADDPG) generally boosts the performance of MADDPG. For the method with population curriculum, vanilla-PC performs almost the same as training from scratch (Att-MADDPG) when the number of agents in the environment is small (i.e., 6-4) but the performance gap becomes much more significant when the population further grows (i.e., 12-18 and 24-16). For our proposed EPC method, it consistently outperforms all the baselines across all the scales. Particularly, in the largest scale 24-16, EPC sheep receive 10x more rewards than the best baseline sheep without curriculum training.

**Detailed Statistics:** Besides rewards, we also compute the statistics of sheep to understand how the trained sheep behave in the game. We perform competitions between sheep trained by different methods against the EPC wolves and measure the average number of total grass pellets eaten per episode, i.e, *#grass eaten*, and the average percentage of sheep that survive until the end of an episode, i.e., *survival rate*, in Fig. 6b. We can observe that as the population increases, it becomes increasingly harder for sheep to survive while EPC trained sheep remain a high survival rate even on the largest scale. Moreover, as more sheep in the game, EPC trained sheep consistently learn to eat more grass even under the strong pressure from wolves. In contrast, the amount of eaten grass of MADDPG approach (i.e., Att-MADDPG) drastically decreases when the number of wolves becomes large.

**Quantitative Results in *Adversarial Battle***

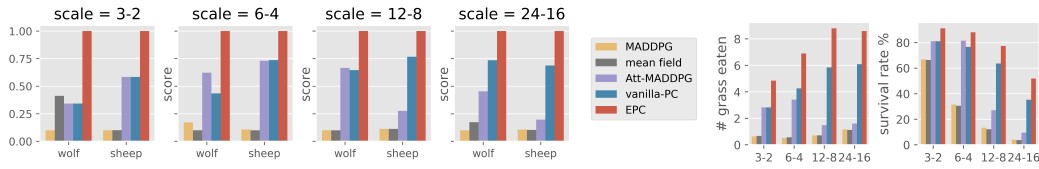

(a) Normalized scores of wolves and sheep

(b) Sheep statistics

Figure 6: Results in *Grassland*. In part (a), we show the normalized scores of wolves and sheep trained by different methods when competing with EPC sheep and EPC wolves respectively. In part (b), we measure the sheep statistics over different scales (x-axis), including the average number of total grass pellets eaten per episode (left) and the average percentage of sheep that survive until the end of episode (right). EPC trained agents (yellow) are consistently better than any baseline method.

In this game, we evaluate on environments with different sizes of agent population $N$, denoted by scale $N_1$-$N_2$ where $N_1 = N_2 = N/2$. We start the curriculum from scale 4-4 and the increase the population size to scale 8-8 ($N = 16$) and finally 16-16 ($N = 32$). Both vanilla-PC and EPC take $5 \times 10^4$ training episodes in the first stage and then $2 \times 10^4$ episodes in the following two curriculum stages. We report the normalized scores of different methods in Fig. 7, where agents trained by EPC outperforms all the baseline methods increasingly more significant as the agent population grows.

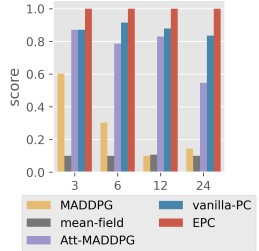

Figure 7: *Adversarial Battle*

**Quantitative Results in *Food Collection***

In this game, we begin curriculum training with $N = 3$, namely 3 agents and 3 food locations, and progressively increase the population size $N$ to 6, 12 and finally 24. Both vanilla-PC and EPC perform training on $5 \times 10^4$ episodes on the first stage of $N = 3$ and then $2 \times 10^4$ episodes in each of the following curriculum stage. We report the normalized scores for all the methods in Fig. 8, where EPC is always the best among all the approaches with a clear margin. Note that the performance of the original MADDPG and the mean-field approach drops drastically as the population size $N$ increases. Particularly, the mean-field approach performs even worse than the

Figure 8: *Food Collection*

original MADDPG method. We believe this is because in this game, the agents must act according to the global team state collaboratively, which means the local approximation assumption in the mean-field approach does not hold clearly.

**Ablative Analysis**

**Stability Analysis:** The evolutionary selection process in EPC not only leads to better final performances but also stabilizes the training procedure. We validate the stability of EPC by computing the variance over 3 training seeds for the same experiment and comparing with the variance of vanilla-PC, which is also obtained from 3 training seeds. Specifically, we pick the second stage of curriculum learning and visualize the variance of agent scores throughout the stage of training. These scores are computed by competing against the final policy trained by EPC. We perform analysis on all the 3 environments: *Grassland* with scale 6-4 (Fig. 9a), *Adversarial Battle* with scale 8-8 (Fig. 9b) and *Food Collection* with scale 6 (Fig. 9c). We can observe that the variance of EPC is much smaller than vanilla-PC in different games.

**Convergence Analysis:** To illustrate that the self-attention based policies trained from a smaller scale is able to well adapt to a larger scale via fine-tuning, we pick a particular mutant by EPC in the second curriculum stage and visualize its learning curve throughout fine-tuning for all the environments, *Grassland* (Fig. 9d), *Adversarial Battle* (Fig. 9e) and *Food Collection* (Fig. 9f). The scores are computed in the same way as the stability analysis. By comparing to MADDPG and Att-MADDPG, which train policies from scratch, we can see that EPC starts learning with a much higher score, continues to improve during fine-tuning and quickly converges to a better solution. Note that all baselines are in fact trained much longer. The full convergence curves are in App. D.1.

**Generalization:** We investigate whether the learned policies can generalize to a different test environment with even a larger scale than the training ones. To do so, we take the best polices trained by different methods on the largest population and directly apply these policies to a new environment with a doubled population by self-cloning. We evaluate in all the environments with EPC, vanilla-PC

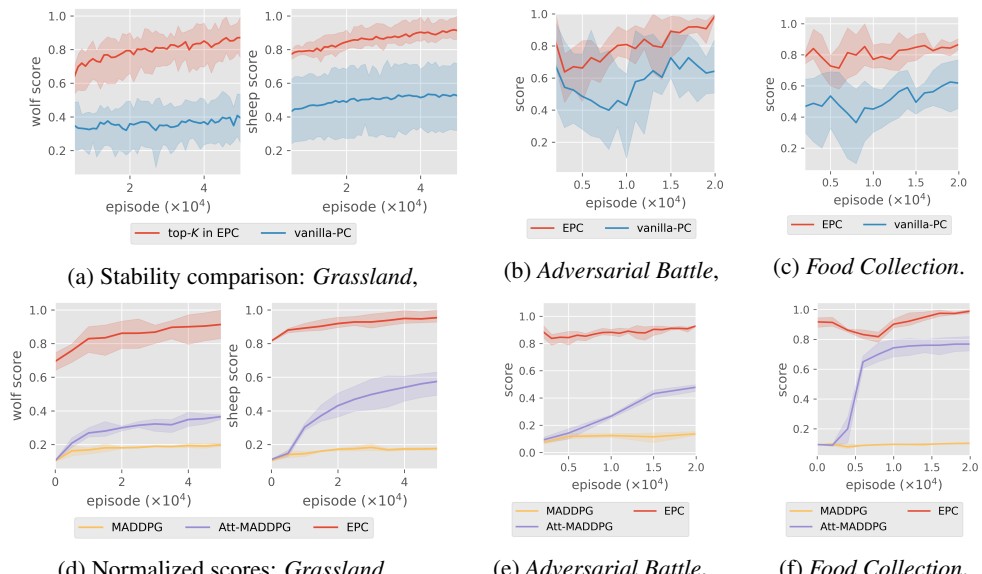

(a) Stability comparison: *Grassland*,  (b) *Adversarial Battle*,  (c) *Food Collection*.

(d) Normalized scores: *Grassland*,  (e) *Adversarial Battle*,  (f) *Food Collection*.

Figure 9: Ablation analysis on the second curriculum stage in all the games over 3 different training seeds. Stability comparison (top) in (a), (b) and (c): We observe EPC has much less variance comparing to vanilla-PC. Normalized scores during fine-tuning (bottom) in (d), (e) and (f): This illustrates that EPC can successfully transfer the agents trained with a smaller population to a larger population by fine-tuning.

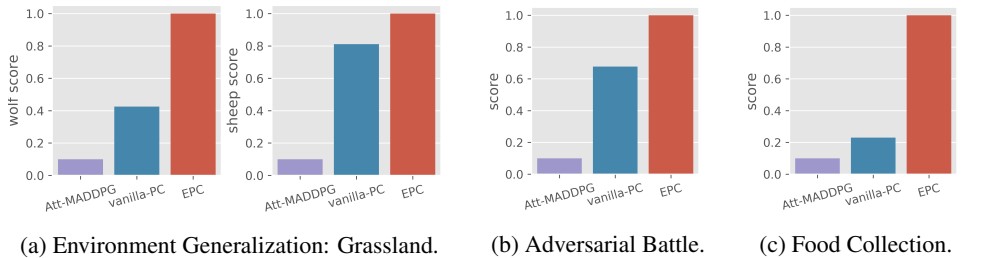

(a) Environment Generalization: Grassland.  (b) Adversarial Battle.  (c) Food Collection.

Figure 10: Environment Generalization: We take the agents trained on the largest scale and test on an environment with twice the population. We perform experiments on all the games and show that EPC also advances the agents' generalizability.

and Att-MADDPG and measure the normalized scores of different methods, which is computed in the same way as the fitness score. In all cases, we observe a large advantage of EPC over the other two methods, indicating the better generalization ability for policies trained by EPC.

## 6    CONCLUSION

In this paper, we propose to scale multi-agent reinforcement learning by using curriculum learning over the agent population with evolutionary selection. Our approach has shown significant improvements over baselines not only in the performance but also the training stability. Given these encouraging results on different environments, we believe our method is general and can potentially benefit scaling other MARL algorithms. We also hope that learning with a large population of agents can also lead to the emergence of swarm intelligence in environments with simple rules in the future.

ACKNOWLEDGMENT

This research is supported in part by ONR MURI N000141612007, ONR Young Investigator to AG. FF is also supported in part by NSF grant IIS-1850477, a research grant from Lockheed Martin, and the U.S. Army Combat Capabilities Development Command Army Research Laboratory Cooperative Agreement Number W911NF-13-2-0045 (ARL Cyber Security CRA). The views and conclusions contained in this document are those of the authors and should not be interpreted as representing the official policies, either expressed or implied, of the funding agencies. We also sincerely thank Bowen Baker and Ingmar Kanitscheider from OpenAI for valuable suggestions and comments.

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

## A   ENVIRONMENT DETAILS

In the *Grassland* game, sheep gets +2 reward when he eats the grass, -5 reward when eaten by wolf. The wolf get +5 reward when eats a sheep. We also shape the reward by distance, sheep will get less negative reward when it is closer to grass and wolf will get less negative reward when it is closer to sheep. This game is adapted from the original *Predator-prey* game in the MADDPG paper (Lowe et al., 2017) by introducing grass and allowing agent to die.

In the *Adversarial Battle* game, agent will get +1 reward when he eats the food, −6 reward when killed by other agents. If $N$ agents kill an enemy, they will be rewarded $+6/N$. We shape the reward by distance. Agent will receive less negative rewards when it is closer to other agents and grass. We want to encourage collision within agents and also will be easier for them to learn to eat. This game is adapted from the mean-field MARL paper (Yang et al., 2018) by converting it from a grid world to particle-world, introducing food and only allowing 2-agent cooperative killing.

In the *Food Collection* game, there are $N$ agents and $N$ food locations. Each agent will get a shared $+6/N$ reward per timestep when one food is occupied by any agent. If one agent gets collision with another, all of the agents will get a punish of $-6/N$. We shape the reward by distance. Agents will receive less negative rewards when it gets closer to the food. Since the number of agents and food are equal, we want to avoid the collision within agents and let the agents to learn to occupy as many food as possible. This is exactly the same game as the *Cooperative Navigation* game in the MADDPG paper. We slightly change the reward function to ensure it is bounded w.r.t. arbitrary number of agents.

We use the *normalized reward* as the score during evaluation. For a particular game with a particular scale, we first collect the reward for each type of agents, namely the average reward of each individual of that type *without* the shaped rewards. Then we re-scale the collected rewards by considering the lowest reward among all methods as score 0 and highest reward as score 1.

## B   TRAINING DETAILS

We follow all the hyper-parameters in the original MADDPG paper (Lowe et al., 2017) for both EPC and all the baseline methods considered. Particularly, we use the Adam optimizer with learning rate 0.01, $\beta_1 = 0.9$, $\beta_2 = 0.999$ and $\varepsilon = 10^{-8}$ across all experiments. $\tau = 0.01$ is set for target network update and $\gamma = 0.95$ is used as discount factor. We also use a replay buffer of size $10^6$ and we update the network parameters after every 100 samples. The batch size is 1024. All the baseline methods are trained for a number of episodes that equals the **accumulative** number of episodes that EPC has taken.

We set $K = 2$ in all the games during training except that $K = 3$ in the food collection game due to computational constraints. During EPC training, in the grassland game, we train the scale of 3 sheep 2 wolf for 100000 episodes. We train another 50000 episodes every time the agents number doubles. In the adversarial battle game and food collection game, we train the first scale for 50000 episodes. We train another 20000 episodes every time the agents number doubles.

In the grassland game, the entity types are the agent itself, other sheep, other wolf and food. We thus have four types of entity encoders for each of those entity types. In the adversarial battle game, Similar to grassland game, the entity types are agent itself, other teammates, enemies and food. We also have four types of entity encoders for each of those entity types. Since there is only one group in the food collection game, the entity types are agent itself, other teammates and food. We thus have three entity encoders in our network.

## C   EVALUATION DETAILS

To evaluate the agents trained in the environment with $\Omega = 2$, we make two roles of agents trained with different approaches compete against each other. Each competition is simulated for 10000 episodes. The average normalized reward over the 10000 episodes will be used as the competition score for each side. Note that in our experiments, we let all the methods compete against our EPC approach for evaluation. For adversarial battle game, we take the average score of two teams as the model's final evaluation score, since the two teams in this game are completely symmetric.

In the food collection game, since there is only one role, we simply simulate the model for 10000 episodes. The average normalized reward over the 10000 episodes will be used as the score of the model.

## D   ADDITIONAL DETAILS ON EXPERIMENT RESULTS

### D.1   FULL TRAINING CURVES FOR BASELINES

All the baseline methods are trained for a number of episodes that equals the **accumulative** number of episodes that EPC has taken.

The purpose of Figure 9e,9e,9f is simply showing the **transfer** performance, i.e., the initialization produced by EPC from the previous stage is effective and can indeed leverage past experiences to warm-start. The x-axis of

the plot was shrunk for visualization purpose. Here we illustrate the complete convergence curve of baselines, i.e., ATT-MADDPG and MADDPG, in Figure 11a,11b,11c for the 3 games respective. Although Att-MADDPG takes a much longer time to learn, its performance is still far worse than EPC.

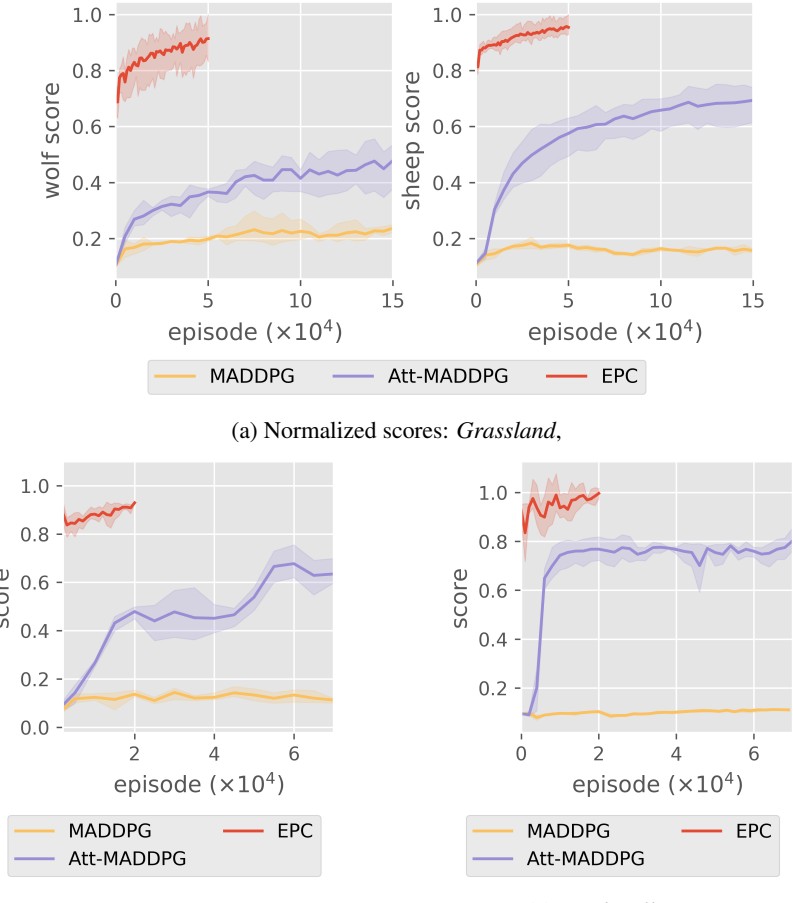

(a) Normalized scores: *Grassland*,

(b) *Adversarial Battle*,

(c) *Food Collection*.

Figure 11: Full learning curves on the second curriculum stage in all the games. EPC fine-tunes the policies obtained from the previous stage while MADDPG and Att-MADDPG are trained from scratch for a much longer time.

## D.2    RAW REWARD NUMBERS OF EVALUATION RESULTS

In this section, we provide the actual rewards without normalization when comparing all the baselines with EPC. These scores are corresponding to the histograms reported in Figure 6a, 7, 8.

*Grassland* game, wolf rewards, corresponding to wolf in Figure 6a:

| scale | MADDPG | mean field | Att-MADDPG | vanilla-PC | EPC |
|---|---|---|---|---|---|
| 3-2 | 0.596 | 0.877 | 0.8145 | 0.8145 | 1.407 |
| 6-4 | 3.7735 | 0.9515 | 2.5905 | 2.001 | 3.7735 |
| 12-8 | 3.1915 | 3.385 | 10.2125 | 9.974 | 14.377 |
| 24-16 | 14.482 | 18.272 | 32.8945 | 47.6365 | 61.4245 |

*Grassland* game, sheep rewards, corresponding to sheep in Figure 6a:

| scale | MADDPG | mean field | Att-MADDPG | vanilla-PC | EPC |
|---|---|---|---|---|---|
| 3-2 | -4.0026 | -3.9947 | 2.66 | 2.66 | 8.3846 |
| 6-4 | -20.2494 | -20.5107 | 0.9892 | 1.1804 | 10.1455 |
| 12-8 | -52.863 | -53.6338 | -42.4801 | -11.3736 | 3.3774 |
| 24-16 | -119.1327 | -118.5668 | -111.0656 | -70.1981 | -44.1031 |

*Adversarial Battle* game, rewards of team 1, corresponding to Figure 7:

| scale | MADDPG | mean field | Att-MADDPG | vanilla-PC | EPC |
|---|---|---|---|---|---|
| 4-4 | 7.51555 | 5.81165 | 22.6357 | 22.6357 | 26.86355 |
| 8-8 | 0.6692 | -0.58115 | 43.7801 | 46.89595 | 65.75585 |
| 16-16 | -46.6398 | -35.5978 | 28.8336 | 109.4406 | 189.69775 |

*Food Collection* game, team rewards, corresponding to Figure 8:

| scale | MADDPG | mean field | Att-MADDPG | vanilla-PC | EPC |
|---|---|---|---|---|---|
| 3 | 55.06 | 42.74 | 61.6488 | 61.6488 | 64.822 |
| 6 | 17.01 | 3.37 | 49.3626 | 58.0014 | 63.7004 |
| 12 | 6.32 | 6.735 | 49.45755 | 52.3625 | 59.54 |
| 24 | 10.346075 | 7.830975 | 33.435 | 49.998025 | 59.47035 |

### D.3  PAIRWISE COMPETITION RESULTS BETWEEN ALL METHODS IN COMPETITIVE GAMES

For visualization purpose, we only illustrate the scores of the competitions between baselines and EPC in the main paper. Here we provide the complete competition rewards between every pair of methods in both *Grassland* and *Adversarial Battle* with the largest population of agents as follows.

Here show the wolf rewards in *Grassland* with scale 24-16. For wolves trained by each approach, we compare them against the sheep by all the methods. EPC wolves always have the highest rewards as in the bottom row. Correspondingly, when different wolves compete against EPC sheep, they always obtain the lowest rewards as in the rightmost column.

| wolf \ sheep | MADDPG | mean-field | Att-MADDPG | Vanilla-PC | EPC |
|---|---|---|---|---|---|
| MADDPG | 66.914 | 67.0945 | 66.34 | 23.048 | 14.482 |
| mean-field | 75.7655 | 74.23 | 74.7375 | 28.3705 | 18.272 |
| Att-MADDPG | 103.22 | 103.326 | 98.07 | 49.557 | 32.8945 |
| Vanilla-PC | 110.333 | 111.3735 | 101.8975 | 64.53 | 47.6365 |
| EPC | 120.9025 | 121.4325 | 115.956 | 82.381 | 61.4245 |

Here show the sheep rewards in *Grassland* with scale 24-16. For sheep trained by each approach, we compete them against the all different wolves. EPC sheep always have the highest rewards as in the last row. Correspondingly, when competing different sheep against EPC wolves, the rewards are always the lowest as in the rightmost column.

| sheep \ wolf | MADDPG | mean-field | Att-MADDPG | Vanilla-PC | EPC |
|---|---|---|---|---|---|
| MADDPG | -63.5096 | -72.5443 | -100.4636 | -107.825 | -119.1327 |
| mean-field | -63.7089 | -71.0714 | -100.6304 | -108.8917 | -118.5668 |
| Att-MADDPG | -62.9416 | -71.3339 | -95.0522 | -99.1207 | -111.0656 |
| Vanilla-PC | -5.7086 | -7.8011 | -31.9936 | -49.5186 | -70.1981 |
| EPC | 9.2135 | 10.5892 | -6.9846 | -27.3475 | -44.1031 |

Here we show the rewards of team 1 in *Adversarial Battle* with scale 16-16. For agents trained by each approach, we compare them as team 1 against all different methods as team 2. When EPC agents as team 1, no matter which opponent is, they always get the highest rewards as in the last row. When other methods compete against EPC, the obtained rewards are always the lowest as in the rightmost column.

| reported \ compared | MADDPG | mean-field | Att-MADDPG | Vanilla-PC | EPC |
|---|---|---|---|---|---|
| MADDPG | 61.4555 | 17.1591 | 2.8033 | -29.9242 | -46.6398 |
| mean-field | 104.41315 | 59.01315 | 27.9004 | 11.1891 | -35.5978 |
| Att-MADDPG | 146.9829 | 117.3804 | 81.702 | 18.57425 | 28.8336 |
| Vanilla-PC | 202.9163 | 155.2805 | 174.91965 | 123.7212 | 109.4406 |
| EPC | 339.63255 | 318.3223 | 256.8464 | 198.3621 | 189.69775 |

### D.4  VARIANCE OF PERFORMANCE EVALUATIONS

We present the performance of all approaches in all three games with the largest scale. We train all the approaches with 3 different seeds and show the normalized scores with variance as following. We can see that EPC not only gives better results but also much smaller variance.

### E  ADDITIONAL EXPERIMENTS ON THE ORIGINAL PREDATOR-PREY GAME

*Grassland* is adapted from the *Predator-prey* game introduced by the original MADDPG paper (Lowe et al., 2017). To further validate our empirical results, we additionally study the performances of different algorithms on the unmodified *Predator-prey* game as follows.

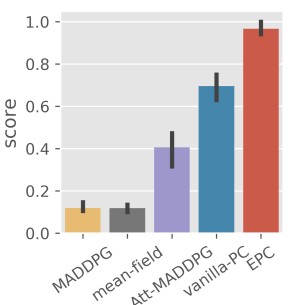 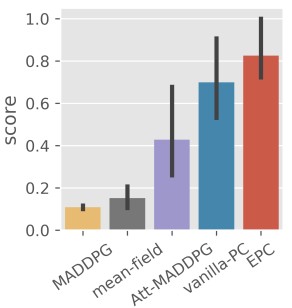 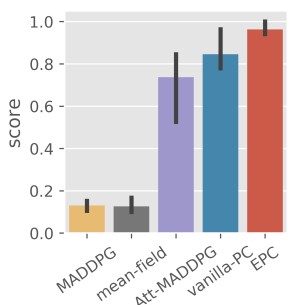

(a) Normalized scores with variances in *Grassland* in scale 24-16

(b) Normalized scores with variances in *Adversarial battle* in scale 16-16

(c) Normalized scores with variances in *Food Collection* with 24 agents

We first report the normalized score in Fig. 13 by comparing all the methods against EPC. EPC is consistently better than all methods in all the scales.

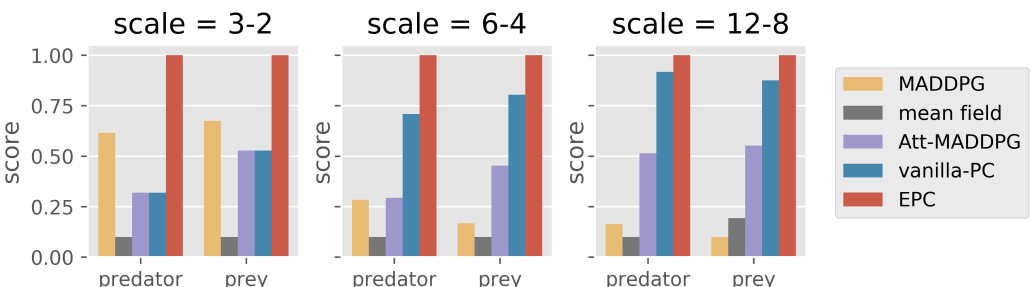

Figure 13: Normalized scores in the original *Predator-prey* game

Besides, we also report the raw reward numbers when competing against EPC. Since the *Predator-prey* game is a zero-sum game, we simply report the predator rewards (the prey reward is exactly the negative value).

| scale | MADDPG | mean field | Att-MADDPG | vanilla-PC | EPC |
|---|---|---|---|---|---|
| 3-2 | 10.405 | 7.39 | 8.675 | 8.675 | 12.662 |
| 6-4 | 31.458 | 25.529 | 31.747 | 45.155 | 54.546 |
| 12-8 | 74.939 | 64.377 | 133.261 | 200.638 | 214.328 |

Furthermore, we also demonstrate the results of full pairwise competition between every two methods for scale 12-8 below. Consistently, we can see that EPC predators always have the highest scores as in the last row. When competing against EPC prey, the lowest rewards are observed.

| predator \ prey | MADDPG | mean-field | Att-MADDPG | Vanilla-PC | EPC |
|---|---|---|---|---|---|
| MADDPG | 229.887 | 249.4 | 247.423 | 122.878 | 74.939 |
| mean-field | 207.022 | 210.838 | 228.469 | 107.811 | 64.377 |
| Att-MADDPG | 569.862 | 532.611 | 373.979 | 204.743 | 133.261 |
| Vanilla-PC | 758.293 | 737.067 | 521.486 | 303.86 | 200.638 |
| EPC | 827.298 | 764.417 | 519.319 | 299.505 | 214.328 |

