# OpenReview forum: "Evolutionary Population Curriculum for Scaling Multi-Agent Reinforcement Learning"
_ICLR.cc/2020/Conference — Accept (Poster)_

### Official Review · AnonReviewer2 · 2019-10-23
**Official Blind Review #2**

**Rating:** 6

**Review:**

Review Update: Thank you for the detailed response, it raised my opinion of the paper as it reduced my concerns on the rigor of the experiments performed. I believe the changes increase the significance of the contribution and may help it reach a broader audience.

--
This paper proposes evolving curriculums of increasing populations of agents to improve multi-agent reinforcement learning with large number of agents. The topic is of relevance to the ICLR community and the results are tending towards publishable contributions, but I have some concerns that I would like the authors to discuss in their rebuttal.

The method is evaluated on a good range of suitably challenging environments. However, why did the authors propose new challenges within the particle environments and not those used in the original publication? This change makes it harder to compare results across publications, whilst not seeming to add a significant change in the learning problem beyond what was present in the original benchmark tasks. The food collection task sounds like it may be equivalent to simple spread. Will the new environments be released as open source for others to build upon this work?

The method is compared against a good range of baseline methods and ablations of the proposed method. However, without grounding the results in a known environment it is hard to place whether the implementations of MADDPG and Mean-Field are fair reproductions. Presuming the authors are using the open source implementation of MADDPG (please confirm) this is most significant for Mean-Field particularly given its poor performance in Section 5.4. Please provide evidence that this is a fair comparison.

To evaluate the resultant agents in the competitive environments, all methods were placed into games against the authors proposed EPC method. Was this the same EPC opponents the evaluated EPC team were trained against? If so, this is an unfair advantage to the EPC team as it has time to optimize against this opponent whilst the other methods have not. Even if it is an EPC team from a different training run, there may be outstanding biases in the joint policies EPC tends towards that benefit the EPC team evaluated. This could be overcome by evaluating all methods in competition with all other methods.

For all experimental results, please quantify variance in performance as well as average value (currently only done in Figure 9 a and b). How many repeats of evaluation and training were performed for each? Without these details the claim on page 9 that "EPC is always the best among all the approaches with a clear margin" is too strong. Are these differences statistically significant? Additionally, please also include the maximum scores (where normalized score = 1.0) for all experiments, as presenting results with only normalized scores unnecessarily reduces the reproducibility of the work.

Finally, in Appendix B, the authors provide a list of hyperparameter settings without discussion of how these were chosen. Were they optimised for one specific method or set to defaults from the literature? In particular, as the performance of Att-MADDPG is still improving at the end of the plot in Figure 9e, I am concerned that the #episodes was chosen to optimise the performance of EPC.

Overall, this is an interesting approach with promising initial results. I believe the contribution would be significantly improved by addressing the issues above and look forward to the authors responses which could increase my rating to acceptance.

Things that could improve the paper but did not impact the score:
1) On page 5 it is noted that the authors do not share parameters between the Q-function and policy. It would improve the paper to justify why this choice was made.
2) Page 5: "N_1 agents for of the role" -> N_1 agents of the role
3) Page 5: "as follows to evolved these K parallel sets" -> evolve
4) Page 7: "resources asThank yoTha green landmarks"
5) Page 8: "understand how the trained sheep behavior in the game" -> how the trained sheep behave in the game
6) Page 14: There are repeated grammatical issues in Appendix A e.g. "is more closer to grass / sheep / other agents" -> is closer to grass / sheep / other agents and "will less negative reward" -> will receive less negative rewards

**Experience Assessment:**

I have published in this field for several years.

**Review Assessment: Checking Correctness Of Derivations And Theory:**

I assessed the sensibility of the derivations and theory.

**Review Assessment: Checking Correctness Of Experiments:**

I carefully checked the experiments.

**Review Assessment: Thoroughness In Paper Reading:**

I read the paper thoroughly.

---

> ### Author Response · Authors · 2019-11-13
> **We have updated our paper with additional experiments and details in Appendix D and E.**
>
> —-”why did the authors propose new challenges ...” “ new environments be released” ”without grounding the results in a known environment it is hard to … are fair reproductions”
> >>> We will release all our source-code. In fact, each of the three environments is either from or slightly adapted from standard benchmarks. We pick two games suitable for many agents from the MADDPG game suites and one from the mean-field MARL paper. We have clarified this in Appendix A in the revision.
>
> The Food Collection game is exactly the same as simple-spread in the original MADDPG paper.
>
> The Grassland Game is a slightly enhanced version of the original predator-prey game in the MADDPG paper, with two enhancements to make it more challenging: (1) agents can die; (2) there are resources.
>
> The Adversarial-Battle game is adapted from the mean-field MARL paper. It is the only high-dimensional experiment in the mean-field paper. The original environment is (1) a grid world and (2) every single agent can kill an enemy (this makes the problem easy since agents need little coordination). We convert it to the particle-world environment with food and further constraint that only two agents can kill an enemy.
>
> Besides, we also add another experiment on the original predator-prey game in Appendix E in the updated paper and all of our conclusions still hold.
>
>
> —- “Please provide evidence that this is a fair comparison”
> >>> Our MADDPG implementation is based on the open-source code of MADDPG from OpenAI.
>
> For mean-field, we have carefully studied their open-source implementation and tried our best to re-implement their algorithm within the MADDPG framework (their code is on Q-learning). We have even obtained confirmation directly from the authors on all our implementation details.
>
> The poor performance of mean-field does not surprise us for the following reasons:
> 1. In their original paper, it states that “We exclude MADDPG as baselines, as the framework of centralized critic cannot deal with the varying number of agents for the battle”. That is, they only compared with single-agent RL baselines but *did NOT compare with MADDPG* at all. We think this comparison can be simply conducted by masking out dead agents (i.e., set them 0). So, we present this missing result in our paper.
> 2. Mean-field takes the average actions of only *nearby* agents within a hand-tuned distance. As we discussed in the introduction, its fundamental assumption is that the Q function can be *linearly approximated by local interactions* (they use Taylor expansion in the proof), which is not guaranteed in many games requiring complex distant coordinations. This is why it performs significantly poor in Food Collection (as we discussed in the texts below fig.8) while on a par with MADDPG in the other two games. Notably, mean-field was not tested in any high-dimensional games other than the grid-world battle game in their original paper.
>
>
> —- “all methods were placed against the EPC method”
> >>> We have included the full pairwise competition results in Appendix D.3 in our revision. We show the rewards by competing against every two methods. This pairwise results show that EPC is consistently better than baselines.
>
> The purpose of the histograms is to show that EPC can “defeat” opponents trained by other methods. The reason for just showing the results against EPC in the main paper is solely for visualization simplicity (since we have 5 methods to compare).
>
>
> —- “please quantify variance ... How many repeats ...”
> >>> For evaluation, as in Appendix C, we run 10000 games to compute each normalized score.
>
> For training, we have included training variance for all methods in fig 9 over 3 seeds (baselines are trained much longer and the full curves are in Appendix D.1).
> For the main results (Fig. 6,7,8), we only did 1 training seed at submission time (similar to the MADDPG paper). Although the training is empirically stable, we agree that it would be better to repeat the training process and include variance. For now, due to compute and time limit, we presented the variance for food collection game in Appendix D.4, which again shows consistent results. We promise to include variance results for all the games in the final version.
>
> We also include the raw reward numbers for Fig. 6,7,8 in Appendix D.2.
>
>
> —- “hyperparameter settings without discussion”
> >>> All hyper-parameters for the MADDPG algorithm are exactly the same as the original MADDPG paper (clarified in Appendix B).
> For the number of iterations, all baselines are trained for a number of episodes that equals the *accumulative* episodes EPC has taken. The purpose of Fig 9(def) is simply showing the *transfer* performance, i.e., the initialization produced by EPC from the previous stage is effective and can indeed leverage past experiences to warm-start. The x-axis of the plot is actually shrunk (Att-MADDPG is trained much longer than one curriculum stage of EPC). We have included the full curves in Appendix D.

---

### Official Review · AnonReviewer1 · 2019-10-23
**Official Blind Review #1**

**Rating:** 8

**Review:**

This paper proposes a new method of scaling multi-agent reinforcement learning to a larger number of agents using evolution. Specifically, the procedures (EPC) involves starting with a small number of agents, training multiple sets in parallel, and doing crossover to find the set of agents that generalize best to a larger number of agents. This is motivated by the intuition that agents that perform best in small groups may not be the ones that perform best in larger groups. These claims are empirically verified in three games based on the particle world set of environments.

I’m a fan of ‘automatic curriculum learning’-style methods designed to gradually add complexity to improve final agent performance, and this paper is no exception. The proposed method is simple, but it makes sense. I like the fact that it is both RL algorithm agnostic, and that it can be largely executed in parallel, which means that it introduces only small training time overhead. I also like the proposed method of making the Q function agnostic to the number of agents and entities using attention (although whether these policies incorporate information from previous time steps, or if they can be made to do so). The experimental results are thorough, comparing to MADDPG, a simpler version of their curriculum without evolution, and a recently proposed method for scaling up MADDPG, showing that EPC consistently outperforms all of them, and is more stable. I think the complexity of the environments is also suitable for this style of paper, although it would be nice to see results in a more open and complex domain such as the recent NeuralMMO game.

Overall, I think this is a good paper and I recommend acceptance.


Small fixes:
-	‘asThank yoTha’ -> not sure what this means
-	N1 agents for of role 1 -> N1 agents of role 1
-	“We adopt the decentralized learning framework” --- even if each agent has their own Q function, if that Q function is centralized (uses the observation of all agents) then training is still centralized


**Experience Assessment:**

I have published one or two papers in this area.

**Review Assessment: Checking Correctness Of Derivations And Theory:**

N/A

**Review Assessment: Checking Correctness Of Experiments:**

I assessed the sensibility of the experiments.

**Review Assessment: Thoroughness In Paper Reading:**

I made a quick assessment of this paper.

---

> ### Author Response · Authors · 2019-11-13
> **Thanks for the valuable comments**
>
> All the typos in the paper have been fixed accordingly.
> We really appreciate the suggestions for testing in more complex environments. NeuralMMO is an excellent option. In the original NeuralMMO paper, even though OpenAI has spent massive compute resources on this project, only maximally 8 individual policies are trained (most of the agents have shared weights).  We are particularly curious to see what will emerge if we can have a much more diverse set of policies deployed in NeuralMMO via EPC. We are working on extending our implementation and plan to apply our work to NeuralMMO in future work.

---

### Official Review · AnonReviewer3 · 2019-10-24
**Official Blind Review #3**

**Rating:** 6

**Review:**

The paper proposes a kind of curriculum for large-scale multi-agent learning. The related work section mentions some obvious points of comparison (note: see also https://science.sciencemag.org/content/364/6443/859.abstract). However, the authors do not compare with ANY of this work (either in terms of algorithm design or performance). It is therefore difficult to evaluate the contribution.

In more detail, the paper combines RL, multi-agent learning and evolution. This is an extremely challenging domain, with many moving parts. How does this approach relate to the work of Salimans et al, Jaderberg et al, Houthooft et al, etc? Without detailed discussion and experiments it is impossible to tell if this is an advance. Improving on the baselines is a useful sanity check. Showing the work is an actual contribution requires comparing against other algorithms in the same space.

---- ----
After reading the rebuttal and other reviews and comments, I've modified my score to weak accept. The paper makes an interesting contribution that is distinct from other approaches.

**Experience Assessment:**

I have published one or two papers in this area.

**Review Assessment: Checking Correctness Of Derivations And Theory:**

I assessed the sensibility of the derivations and theory.

**Review Assessment: Checking Correctness Of Experiments:**

I assessed the sensibility of the experiments.

**Review Assessment: Thoroughness In Paper Reading:**

I read the paper thoroughly.

---

> ### Author Response · Authors · 2019-11-13
> **We tackle a completely different problem from those cited evolution-in-RL references**
>
> We don’t think our work can be directly compared against the references in the “evolution” paragraph of the related work section since the evolution in EPC is addressing a *different problem*.
>
> It is a continuous trend of applying evolution to solve a variety of different challenges in RL and our paper simply tackles a novel one. Our primary focus is to propose a general curriculum learning paradigm to effectively scale MARL to a larger number of agents while the use of evolution is to address a particular *objective misalignment issue* within this paradigm, which has not yet been studied in the literature to our best knowledge. We have compared with the papers which also study scaling MARL while it is out of the scope of our paper to investigate every parallel technique in the domain.
>
> We have updated the related work section to better express the above message.
>
> To be more clear about the differences between the problem we studied and those cited related works, here are detailed summarization/discussion of each paper we mentioned:
> > Salimans et al. (2017) show that population-based training can be better parallelized than standard RL algorithms. It focuses on single-agent RL benchmarks and suggests evolution can be an alternative to PG.
> > Jaderberg et al. (2017) is the first population-based training paper from Google which boosts the performance of many benchmark tasks by running evolution on hyper-parameter tuning.
> > Houthooft et al. (2018) propose to directly run evolution to learn a neural loss function to replace the PG loss, which is extremely expensive, but experiments on small single-agent tasks show that policies learn by the evolved loss can generalize better.
> > Khadka et al. (2018) run population-based training and off-policy RL training together to leverage the diverse samples collected by the population to improve off-policy learning.
> > Conti et al. (2018) proposed an improved exploration technique for running evolution algorithm in RL.
>
> Thanks for mentioning the Capture-The-Flag paper (Jaderberg et al. 2018) from Google. It uses exactly the same training framework from Jaderberg et al. (2017) to solve the game. Particularly, 30 agents are trained as a population and the evolution algorithm is performed to tune each agent’s intrinsic reward to overcome the sparse success reward of the game.  While in our work, we use evolution to tackle the objective misalignment challenge when the number of agents is increased. These are two very different problems of interest. We have cited this paper in the revision. Notably, even in this mentioned paper, there is no direct comparison with any evolution references. Instead, the proposed method is compared with only two baselines, i.e., (1) pure self-play + sparse reward and (2) self-pay + hand-tuned intrinsic rewards.

---

### Author Response · Authors · 2019-11-13
**We have updated our paper with changes in red color**

We have made the following revisions to our paper with all the changes colored in red. We promise to release all the code in the final version.
1. New Appendix D: We add more evaluation details and additional experimental results.
2. New Appendix E: Since our Grassland game is adapted from the original zero-sum Predator Prey game from the MADDPG paper, we conduct additional experiments on the unmodified predator prey game to validate our implementations.
3. Fig.9 updated: we visualize training variances for all baselines.
4. Related work section updated: In the “evolutionary learning” paragraph, we put more details of existing literature.
5. We clarify that all baselines are trained with the same accumulative episodes as EPC took.
6. More details in Appendix A & B.
7. All typos are fixed

---

### Decision · Program_Chairs · 2019-12-19

**Decision:**

Accept (Poster)

**Comment:**

The paper proposes a curriculum approach to increasing the number of agents (and hence complexity) in MARL.

The reviewers mostly agreed that this is a simple and useful idea to the MARL community. There was some initial disagreement about relationships with other RL + evolution approaches, but it got resolved in the rebuttal. Another concern was the slight differences in the environments considered by the paper compared to the literature, but the authors added an experiment with the unmodified version.

Given the positive assessment and the successful rebuttal, I recommend acceptance.